# Optical–Mechanical Integration Analysis and Validation of LiDAR Integrated Systems with a Small Field of View and High Repetition Frequency

Lu Li [1,2], Kunming Xing [2], Ming Zhao [2,3,*], Bangxin Wang [2], Jianfeng Chen [2,4] and Peng Zhuang [5]

1   School of Mechanical and Automotive Engineer, West Anhui University, Lu'an 237012, China; lilu1225@mail.ustc.edu.cn
2   Key Laboratory of Atmospheric Optics, Anhui Institute of Optics and Fine Mechanics, Chinese Academy of Sciences, Hefei 230031, China; kunmingx@mail.ustc.edu.cn (K.X.); wyj@aiofm.ac.cn (B.W.); jianfengchen1212@mail.ustc.edu.cn (J.C.)
3   School of Electronic Engineering, Huainan Normal University, Huainan 232038, China
4   Science Island Branch of Graduate School, University of Science and Technology of China, Hefei 230026, China
5   Anhui Lanke Information Technology Co., Ltd., Hefei 230088, China; pzhuang007@163.com
*   Correspondence: zhaom@aiofm.ac.cn; Tel.:+86-135-1566-7855

**Abstract:** Integrated systems are facing complex and changing environments with the wide application of atmospheric LiDAR in civil, aerospace, and military fields. Traditional analysis methods employ optical software to evaluate the optical performance of integrated systems, and cannot comprehensively consider the influence of optical and mechanical coupling on the optical performance of the integrated system, resulting in the unsatisfactory accuracy of the analysis results. Optical–mechanical integration technology provides a promising solution to this problem. A small-field-of-view LiDAR system with high repetition frequency, low energy, and single-photon detection technology was taken as an example in this study, and the Zernike polynomial fitting algorithm was programmed to enable transmission between optical and mechanical data. Optical–mechanical integration technology was employed to obtain the optical parameters of the integrated system under a gravity load in the process of designing the optical–mechanical structure of the integrated system. The experimental validation results revealed that the optical–mechanical integration analysis of the divergence angle of the transmission unit resulted in an error of 2.586%. The focal length of the telescope increased by 89 μm, its field of view was 244 μrad, and the error of the detector target surface spot was 4.196%. The continuous day/night detection results showed that the system could accurately detect the temporal and spatial variations in clouds and aerosols. The inverted optical depths were experimentally compared with those obtained using a solar photometer. The average optical depth was 0.314, as detected using LiDAR, and 0.329, as detected by the sun photometer, with an average detection error of 4.559%. Therefore, optical–mechanical integration analysis can effectively improve the stability of the structure of highly integrated and complex optical systems.

**Keywords:** LiDAR; integrated system; optical–mechanical integration analysis; Zernike polynomial; clouds and aerosols; optical depth

## 1. Introduction

LiDAR system structure design involves the comprehensive application of various aspects, including optical, mechanical, electrical, thermal, and software design. The continuous development of laser and detection technologies, particularly the ongoing maturation of miniature laser technology, has enabled the effective control of LiDAR integrated systems in terms of size and cost [1,2]. Typical varieties of micro-pulse LiDAR (MPL), characterized by a high repetition rate and low energy, have been widely studied and applied. The Global Micro-Pulse LiDAR Network (MPLNET), established by the National Aeronautics and

Space Organization of the United States, is used for full-time, fully automatic observation of aerosols and clouds worldwide [3,4]. The MiniMPL, produced by Sigma Space, has a detection range of about 10 km during the day; at night, the detection distance is about 20 km [5]. The Cloud Aerosol Transport System (CATS) [6,7] was developed by NASA in the United States and has an orbital altitude of 405 km. Two launch systems were designed to achieve different detection modes. One of the lasers was inherited and developed from Cloud Physics LiDAR (CPL) [8], with a repetition rate of 5 kHz and an energy of 2 mJ; the other laser features a seed injection pulse laser, which was inherited and developed with the airborne cloud–aerosol transmission system [9], with a repetition frequency of 4 kHz and an energy of 2 mJ. MPL has been developed by relevant scientific research institutions in China for vertical, horizontal, and three-dimensional scanning detection of atmospheric aerosols and clouds. According to the detection results, the detection distance is 3–5 km during the day, and it is 10–15 km at night [10–12].

The single-photon detection mode was adopted for the acquisition system of a LiDAR with a high repetition frequency and low-energy laser, utilizing a small field of view to suppress the background signals during the daytime, which requires higher structural stability of the integrated system. The system serves as a data validation system for the development of spaceborne LiDAR in the later stage. Currently, in the published research literature on optical–mechanical–thermal integration analysis of integrated systems, there has been no systematic analysis, and studies have neglected the influence of the structural micro-vibration of optical elements inside the optical system on the optical quality of the atmospheric LiDAR integrated systems. Although specialized analysis and design software are available for optical, mechanical, and thermal disciplines, they are unable to deal with dynamic interdisciplinary problems in the overall design and analysis of systems. Optical–mechanical–thermal integration analysis, a comprehensive interdisciplinary analysis technique that combines the disciplines of optics, structure, and thermology [13,14], was utilized as shown in Figure 1.

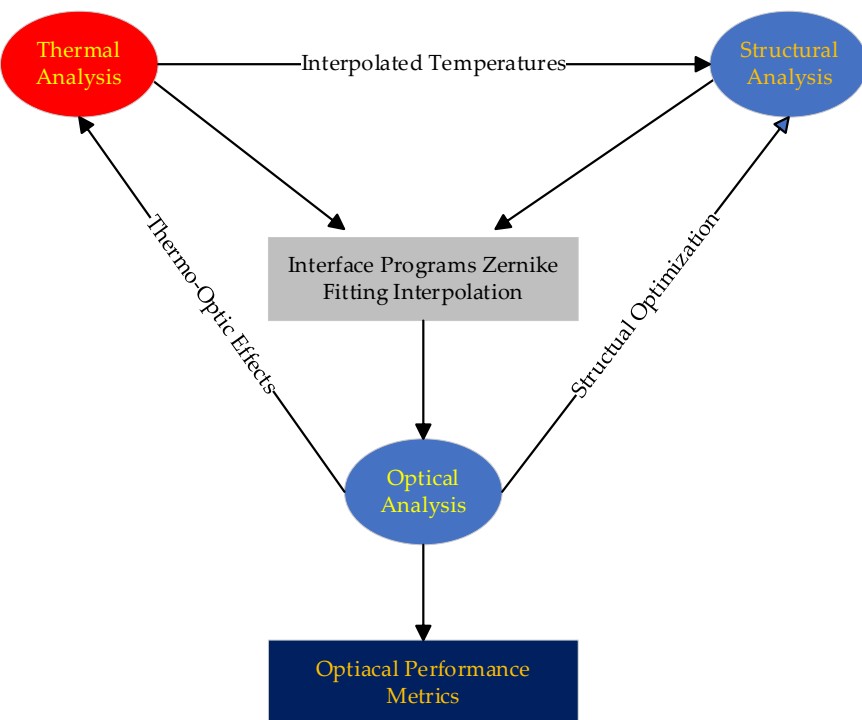

**Figure 1.** Optical–mechanical–thermal integrated analysis.

To date, this method has been applied to the design of optical instruments such as space optical instruments and large ground-based telescopes, and it is used to evaluate whether the optical performance of optical systems fulfills the requirements under the action

of mechanical/thermal environmental loads [15–20]. The characteristics of the thermal–optical–mechanical elements in the Subarcsecond Telescope and BaLloon Experiment (STABLE) directly affect the quality of the point-spread function of the guide star on the detector, so a series of thermal, structural, and optical models were built to simulate system performance and ultimately inform the final pointing stability predictions [15]. During the development process of projects such as the Laser Interferometer Space Antenna (LISA) [16], Wide-Field Infrared Survey Telescope (WFIRST) [17], Narrow-Field Infrared Adaptive Optics System (NFIRAOS) [18], 4 m telescope for habitable exoplanet observation mission (HabEx) [19], and Advanced Terrain Laser Altimeter System (ATLAS) [20] in the ICESat-2 mission, an optical mechanical thermal integration analysis model was established, and the analysis results were validated for the system design.

In this study, a small-field-of-view LiDAR system with high repetition frequency, low energy, and single-photon detection technology was considered; an optical–mechanical integrated analysis model was established; the impact of gravity on the optical performance of the integrated system was investigated; and the feasibility of the method was verified experimentally. Through this study, technical support is provided for further research on the impact of temperature changes on the optical performance of the atmospheric LiDAR integrated systems and the optimization of the optical–mechanical structure design of the spaceborne LiDAR system in the later stage.

## 2. Technological Principles of Conversion between Optical and Mechanical Data

In the finite element analysis of the structural mechanics of the LiDAR optical–mechanical system design with a small field of view and high repetition frequency, the results obtained are usually model node displacements, stresses, or strains, which cannot be used directly to analyze the performance of the optical system and must be converted into optical surface motion data. Optical surface deformation data usually include rigid-body displacements and surface deflections [21]. There are three main forms of rigid-body displacement, namely, defocusing, eccentricity, and tilting, which directly lead to changes in the relative positions between the surfaces of optical components in the optical system and may result in performance degradation or even failure of the optical–mechanical system. For a small-field-of-view, high-repetition-frequency LiDAR, its transmission, receiving, and aft optics have strict requirements on the relative positions between the optical components. Surface deflections are mainly caused by internal stresses, affecting the optical performance of the optical system.

### 2.1. Calculation of Rigid-Body Displacement

Under the effect of mechanical or thermal loads, the rigid-body displacement of an optical element surface can be decomposed into translation and rotation values along the three directions of the coordinate system [22]. $U_x$, $U_y$, and $U_z$ are set to the translation values of the optical surface along the three axes; $\theta_x$, $\theta_y$, and $\theta_z$ are the rotation values of the optical surface along the three axes; $(x_i, y_i, z_i)^T$ and $(x'_i, \ y'_i, \ z'_i)^T$ are the coordinates of the optical surface node before and after deformation, respectively; and $\Delta x_i$, $\Delta y_i$, $\Delta z_i$ are the displacements generated by the deflection of the optical surface. According to the homogeneous coordinate transformation theory, the homogeneous coordinate transformation matrix can be deduced as follows [23,24]:

$$A = \begin{bmatrix} 1 & -\theta_z & \theta_y & u_x \\ \theta_z & 1 & -\theta_x & u_y \\ -\theta_y & \theta_x & 1 & u_z \\ 0 & 0 & 0 & 1 \end{bmatrix} \tag{1}$$

The coordinate relationship of each node before and after the deformation of the optical surface is established as follows:

$$
\begin{bmatrix} x_i' \\ y_i' \\ z_i' \\ 1 \end{bmatrix} = \begin{bmatrix} 1 & -\theta_z & \theta_y & u_x \\ \theta_z & 1 & -\theta_x & u_y \\ -\theta_y & \theta_x & 1 & u_z \\ 0 & 0 & 0 & 1 \end{bmatrix} \times \begin{bmatrix} x_i \\ y_i \\ z_i \\ 1 \end{bmatrix} + \begin{bmatrix} \Delta x_i \\ \Delta y_i \\ \Delta z_i \\ 0 \end{bmatrix} \tag{2}
$$

The results are deduced as follows:

$$
\begin{cases} \Delta x_i = x_i' - x_i + \theta_z y_i - \theta_y z_i - u_x \\ \Delta y_i = y_i' - y_i - \theta_z x_i + \theta_x z_i - u_y \\ \Delta z_i = z_i' - z_i + \theta_y x_i - \theta_x y_i - u_z \end{cases} \tag{3}
$$

According to the least-squares method, the evaluation function is established as

$$
E = \sum_i w_i \left[ \left( x_i' - x_i + \theta_z y_i - \theta_y z_i - u_x \right)^2 + \left( y_i' - y_i - \theta_z x_i + \theta_x z_i - u_y \right)^2 + \left( z_i' - z_i + \theta_y x_i - \theta_x y_i - u_z \right)^2 \right] \tag{4}
$$

$W_i$ is the weighting factor of the $i$-th displacement point of the surface shape. The partial derivative of $E$ is taken for each term of the rigid-body displacements, and the values are set to 0:

$$
\frac{\partial E}{\partial u_x} = \frac{\partial E}{\partial u_y} = \frac{\partial E}{\partial u_z} = \frac{\partial E}{\partial \theta_x} = \frac{\partial E}{\partial \theta_y} = \frac{\partial E}{\partial \theta_z} = 0 \tag{5}
$$

The matrix form is derived as

$$
\begin{bmatrix} u_x \\ u_y \\ u_y \\ \theta_x \\ \theta_y \\ \theta_z \end{bmatrix} = \begin{bmatrix} n & 0 & 0 & 0 & \sum_{i=1}^n z_i & -\sum_{i=1}^n y_i \\ 0 & n & 0 & -\sum_{i=1}^n z_i & 0 & \sum_{i=1}^n x_i \\ 0 & 0 & n & \sum_{i=1}^n y_i & -\sum_{i=1}^n x_i & 0 \\ \sum_{i=1}^n y_i & -\sum_{i=1}^n x_i & 0 & \sum_{i=1}^n x_i z_i & \sum_{i=1}^n y_i z_i & -\sum_{i=1}^n (x_i^2 + y_i^2) \\ -\sum_{i=1}^n z_i & 0 & \sum_{i=1}^n x_i & \sum_{i=1}^n x_i y_i & -\sum_{i=1}^n (x_i^2 + z_i^2) & \sum_{i=1}^n y_i z_i \\ 0 & \sum_{i=1}^n z_i & -\sum_{i=1}^n y_i & -\sum_{i=1}^n (y_i^2 + z_i^2) & \sum_{i=1}^n x_i y_i & \sum_{i=1}^n x_i z_i \end{bmatrix}^{-1} \begin{bmatrix} \sum_{i=1}^n (x_i' - x_i) \\ \sum_{i=1}^n (y_i' - y_i) \\ \sum_{i=1}^n (z_i' - z_i) \\ \sum_{i=1}^n [y_i(x_i' - x_i) - x_i(y_i' - y_i)] \\ \sum_{i=1}^n [x_i(z_i' - z_i) - z_i(x_i' - x_i)] \\ \sum_{i=1}^n [z_i(y_i' - y_i) - y_i(z_i' - z_i)] \end{bmatrix} \tag{6}
$$

To verify the calculation accuracy of the derived rigid-body displacement separation formula, a set of rigid-body displacement covariates was established, random numbers were introduced to simulate wavefront distortion, and the coordinates after deformation were calculated using Equation (2). Finally, the rigid-body displacement separation values were calculated using Equation (6) and compared with the set values as given in Table 1. The distortion was set to $0.01 \times (-1)^{(i)} \times R$, where $R$ produces random numbers from 0 to 1 and $(-1)^{(i)}$ refers to determining whether the representing distortion of the $i$-th displacement point is positive or negative.

**Table 1.** Computational accuracy of rigid-body displacement separation.

| Parameter | Set Value | Calculated Value |
|---|---|---|
| $U_x$ | 1 mm | 0.999978 mm |
| $U_y$ | 2 mm | 1.999996 mm |
| $U_z$ | 3 mm | 3.000007 mm |
| $\theta_x$ | 0.1 mrad | 0.100341 mm |
| $\theta_y$ | 0.1 mrad | 0.100111 mm |
| $\theta_z$ | 0.1 mrad | 0.09980 mm |

According to Table 1, the calculation error of the rigid-body displacement using Equation (6) does not exceed 0.35%, which meets the calculation requirements.

*2.2. Polynomial Fitting of the Optical Surface*

When optical instruments are subjected to mechanical or thermal loads, surface deflection will occur on the optical surface of the components, having a certain impact on their performance [25,26]. Therefore, when designing high-precision optical instruments, the deflections of the optical surfaces of the components must be analyzed and verified for compliance with the design specifications (root mean square and peak to valley as design specifications) using inspection techniques. As for optical surface fitting, the Zernike polynomials are generally utilized to fit surface deflections, and the wavefront polynomials are obtained as follows:

$$
\begin{cases}
W(r) = \sum\limits_{i=0}^{k} A_i Z_i(r) = \sum\limits_{n}^{k} \sum\limits_{m=0}^{n} a_n^m Z_n^m(\rho, \theta) \\
R_n^m(r) = \sum\limits_{s=0}^{\frac{n-m}{2}} (-1)^s \dfrac{(n-s)!}{s!\left(\frac{n+m}{2}-s\right)!\left(\frac{n-m}{2}-s\right)!} r^{(n-2s)} \\
Z_n^m(\rho, \theta) = \sqrt{2(n+1)} R_n^m(\rho) \begin{cases} \cos m\theta \\ \sin m\theta \end{cases}
\end{cases}
\tag{7}
$$

In the above equation, $A_i$ is the Zernike polynomial coefficient. To solve the Zernike polynomial coefficient, the least-squares method was used. The entire fitting process was carried out on a unit circle with normalized coordinate values, and the wavefront polynomial was rewritten as follows:

$$
W_i(r_i) = \sum_{j=0}^{N} A_j Z_j(r_i)
\tag{8}
$$

In the above equation, $W_i(r_i)$ is the wavefront value of node $i$, $Z_{ij} = Z_j(r_i)$, and $i = 1$, 2, ..., $M$, where $M$ is the sum of all the nodes on the optical surface; and $j = 1, 2, ..., N$, where $N$ is the number of Zernike polynomial terms. The expansion is as follows:

$$
\begin{cases}
Z_{11} A_1 + Z_{12} A_2 + \cdots + Z_{1n} A_n = W_1 \\
Z_{21} A_1 + Z_{22} A_2 + \cdots + Z_{2n} A_n = W_2 \\
\qquad\qquad\qquad \vdots \\
Z_{m1} A_1 + Z_{m2} A_2 + \cdots + Z_{mn} A_n = W_m
\end{cases}
\tag{9}
$$

Using the least-squares method, Equation (9) satisfies the following relationship:

$$
\Delta^2 = \sum_{i=1}^{M} \left[ \sum_{j=0}^{N} A_i Z_i(r_j) - W_i(r_i) \right]^2
\tag{10}
$$

The following equation is then deduced by taking the partial derivatives of $\Delta^2$ for each of $A_1, A_2, \cdots, A_k$ and setting the values to 0:

$$
\frac{\partial}{\partial A_k}\left(\Delta^2\right) = 2\sum_{i=1}^{M}\left[\sum_{j=0}^{N} A_j Z_j(r_i) Z_k(r_j) - W_i(r_i) Z_k(r_j)\right] = 2\sum_{i=1}^{M}\left[\sum_{j=0}^{N} A_j Z_j(r_i) Z_k(r_j) - 2\sum_{j=0}^{N} W_i(r_i) Z_k(r_j)\right] = 0
\tag{11}
$$

The above equation can be converted into

$$
\sum_{j=0}^{N}\left[\sum_{i=1}^{M} Z_j(r_i) Z_k(r_j)\right] A_k = \sum_{j=0}^{N} W_i(r_i) Z_k(r_j)
\tag{12}
$$

The above equation is a least-squares canonical equation, with $k = 1, 2, ..., N$.

### 3. Optical–Mechanical Integration Analysis of Small-Field-of-View, High-Repetition-Frequency LiDAR

LiDAR integrated systems with a small field of view and high repetition frequency ware developed by the Key Laboratory of Atmospheric Optics Center of Anhui Institute of Optics and Mechanics, Chinese Academy of Sciences. Figure 2 shows the principle and internal structure of the system [2]. The transmission unit emits a 532.18 nm laser via the Nd:YAG high-repetition-frequency pulse laser. The direction of the laser is adjusted by the reflecting mirror. The laser divergence angle is compressed by the 20× beam expander. Next, the laser is emitted into the atmosphere. The backscattered echo signal is received by the telescope. The hole diameter of the iris is 0.4 mm, which limits the receiving field of view to reduce the background noise. The ocular further converts the echo signal into parallel light. Then, the parallel light is adjusted by the reflecting mirror toward the polarizing prism, so that it is divided into horizontal and vertical detection channels. Finally, the parallel light is filtered by the 0.3 nm narrowband interference filter and focused onto the target surface of the PMT by the lens.

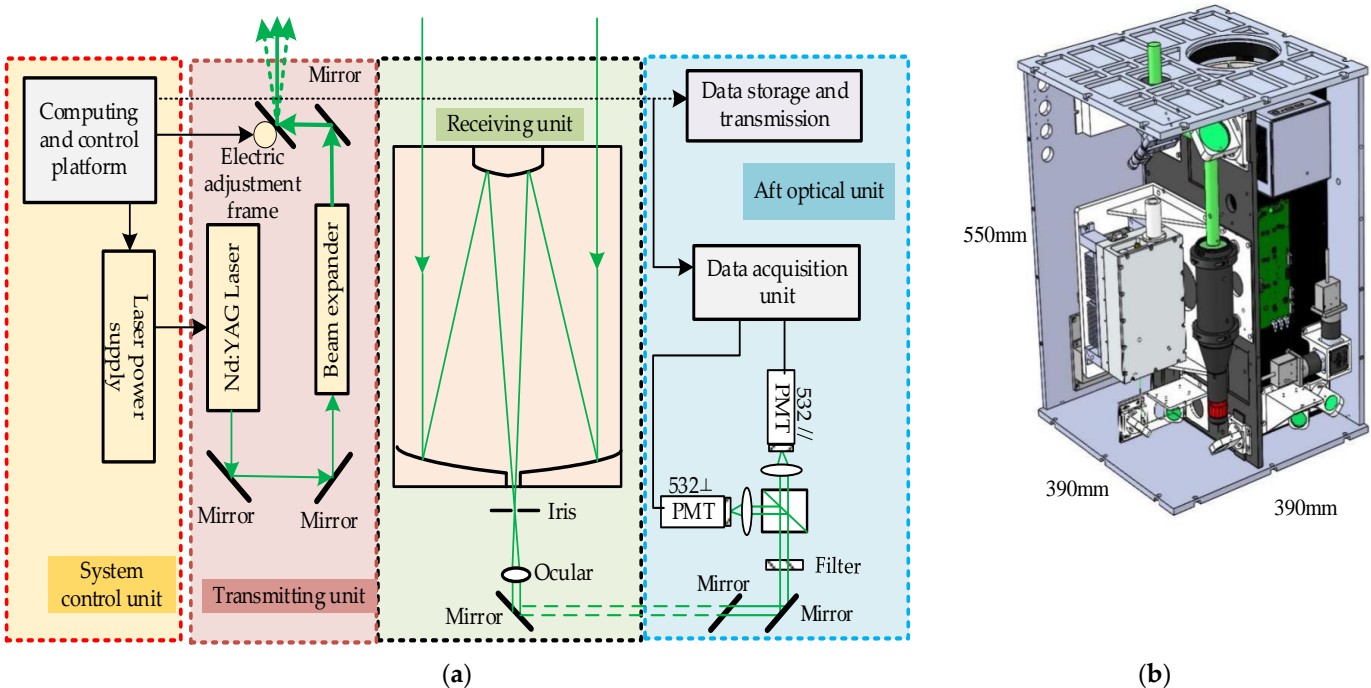

**Figure 2.** LiDAR integrated systems: (**a**) schematic diagram and (**b**) internal structure diagram.

The 2-inch reflecting mirror at the last stage of the transmission unit is installed on the picomotor piezoelectric reflector adjustment frame. When the LiDAR system is subjected to external loads that cause the direction of the emitted laser beam to deviate from the receiving field of view, the control system can adjust the reflection angle to make the receiving and transmission paths parallel. The transmission, receiving, and aft optical path systems were simulated using optical design software, and the divergence angle of the transmission system and the receiving field of view were calculated to be 106 and 250 µrad, respectively, without considering the gravity load. The system parameters of LiDAR are shown in Table 2.

**Table 2.** System parameters of LiDAR.

| Item | Parameters | Value |
|---|---|---|
| Transmission unit | Wavelength | 532.18 nm |
| | Repetition Rate | 3 kHz |
| | Output Divergence | 106 μrad (full) (without considering the gravity load) |
| | Output beam energy | 1 mJ |
| Receiving and aft optical units | Telescope diameter | 125 mm |
| | Iris | 0.4 mm |
| | Field of view | 250 μrad (without considering the gravity load) |
| | Telescope Focal length | 1430 nm |
| | Filter bandwidth | 0.3 nm |
| | Focal length of ocular | 50 mm |
| | Reflector (532 nm) | R:99% |
| | Extinction ratio of polarizing prism | 3000:1 |
| Data acquisition and LiDAR control units | Detector | PMT |
| | Acquisition mode | Photon-counting |
| | Number of Channels | 2 |
| | Data storage mode | Storage or Sending |

### 3.1. Finite Element Model and Local Coordinate System Establishment of the Integrated LiDAR System

To effectively construct the finite element model of the integrated system, the system structure must be simplified, as shown in Figure 3a. In the simplified structure, the laser is mass point A, with a mass of 5 kg acting on the fixed aperture, the motorized adjusting structure and its fixed lens denote mass point B, with a mass of 0.201 kg acting on the fixed aperture, the beam expanding system is mass point C, with a mass of 0.500 kg acting on the fixed aperture, the polarizing prism and detector represent mass point D, with a mass of 0.200 kg acting on the fixed prism structure, the industrial personal computer and its fixed structural parts are mass point E with a mass of 0.200 kg acting on the fixed aperture, the acquisition card and its fixed structural parts are mass point F with a mass of 0.200 kg acting on the fixed aperture, the electronic control unit is mass point G, with a mass of 5 kg acting on the fixed plate surface, and the high-precision three-dimensional adjusting structure is mass point H, with a mass of 0.500 kg acting on the fixed aperture.

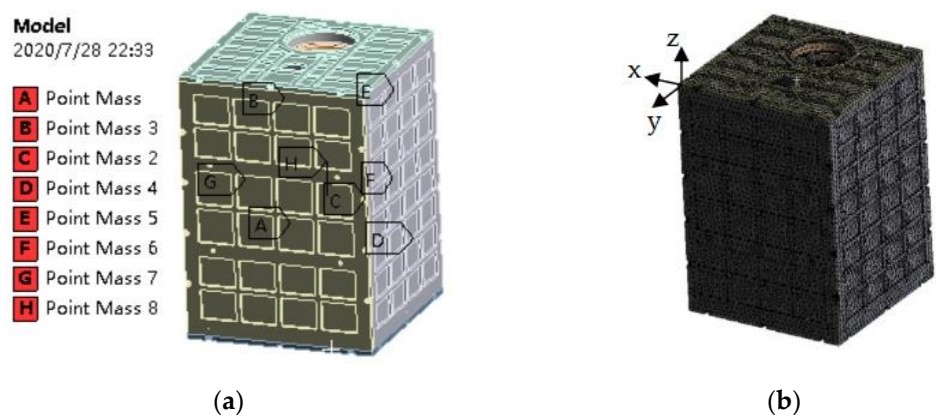

| | |
|---|---|
| (**a**) | (**b**) |

**Figure 3.** Three-dimensional and finite element models of the LiDAR system: (**a**) three-dimensional simplified model; (**b**) finite element model.

Figure 3b depicts the finite element model of the integrated system, which contains 3,011,735 nodes and 1,910,409 grid cells. Optical–mechanical integration analysis of the integrated system was performed as described in the following.

Because the coordinate system of the simulation software is the global coordinate system of the structure and the coordinate system used for rigid-body displacement and polynomial fitting is the local coordinate system established on the reflecting surfaces, a

local coordinate system was established for each reflecting surface before conducting the corresponding rigid-body displacement separation and polynomial fitting as shown in Figure 4.

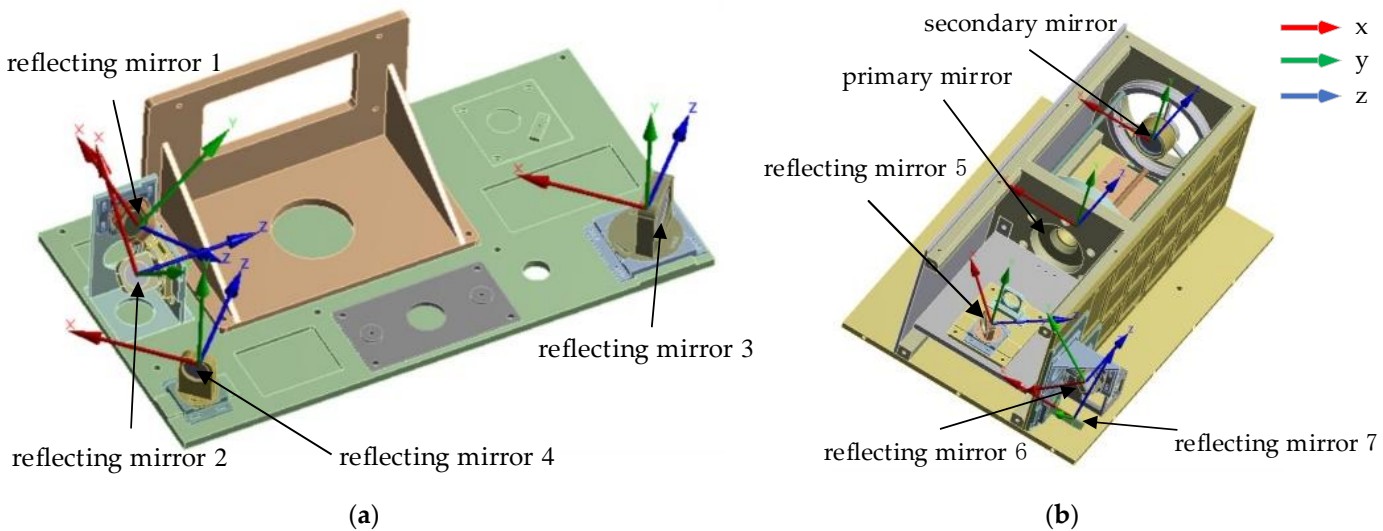

**Figure 4.** Establishment of local coordinate systems for each mirror of the transmitting unit: (**a**) transmitting unit; (**b**) receiving and aft optical units.

### 3.2. Optical Element Deformation Analysis

The Y direction of the gravity load is the placement direction for the installation and testing of the LiDAR system; therefore, the analysis was performed with the Y direction of the gravity load to obtain the deformation of the integrated system under the effect of gravity. Further, the effect of the deformation of each optical element on the optical performance of the integrated system was determined via optical–mechanical integration analysis.

Figure 5 depicts the cloud diagram of reflecting mirror deformation under a gravity load. According to the results, the maximum value of deformation, 1.1177 μm, which was recorded for reflecting mirror 1, was utilized as shown in Figure 5a. The maximum value of deformation, 2.8406 μm, which was recorded for reflecting mirror 5, was utilized as shown in Figure 5b. With the corresponding local coordinate system established, the rigid-body displacement of each reflecting surface was calculated using the rigid-body displacement separation technique using the displacement data of each beam-folding mirror surface. The results are presented in Table 3.

**Table 3.** Simulation results of the corresponding local coordinate system of optical elements under a gravity load.

| Item | Translation (μm) | | | Tilt Angle (°) | | |
|---|---|---|---|---|---|---|
| | X | Y | Z | X | Y | Z |
| Reflecting mirror 1 | $-2.151 \times 10^{-1}$ | $1.672 \times 10^{-1}$ | $9.094 \times 10^{-1}$ | $5.229 \times 10^{-6}$ | $3.408 \times 10^{-5}$ | $2.805 \times 10^{-6}$ |
| Reflecting mirror 2 | $-8.779 \times 10^{-2}$ | $-2.288 \times 10^{-1}$ | $2.592 \times 10^{-1}$ | $-3.328 \times 10^{-6}$ | $4.259 \times 10^{-6}$ | $-6.560 \times 10^{-6}$ |
| Reflecting mirror 3 | $2.191 \times 10^{-3}$ | $-4.060 \times 10^{-2}$ | $-2.403 \times 10^{-2}$ | $-2.259 \times 10^{-7}$ | $1.334 \times 10^{-6}$ | $-5.881 \times 10^{-8}$ |
| Reflecting mirror 4 | $4.467 \times 10^{-3}$ | $-4.368 \times 10^{-2}$ | $-5.510 \times 10^{-3}$ | $5.175 \times 10^{-8}$ | $-3.992 \times 10^{-9}$ | $4.031 \times 10^{-8}$ |
| Primary mirror | $-6.161 \times 10^{-4}$ | $-1.565 \times 10^{-1}$ | $-3.184 \times 10^{-2}$ | $-2.444 \times 10^{-7}$ | $1.543 \times 10^{-8}$ | $5.045 \times 10^{-9}$ |
| Secondary mirror | $-1.180 \times 10^{-3}$ | $-9.007 \times 10^{-1}$ | $-4.809 \times 10^{-2}$ | $-2.737 \times 10^{-6}$ | $3.182 \times 10^{-8}$ | $-1.118 \times 10^{-9}$ |
| Reflecting mirror 5 | $-5.118 \times 10^{-1}$ | $-2.460 \times 10^{-3}$ | $-5.345 \times 10^{-1}$ | $-1.438 \times 10^{-5}$ | $-2.527 \times 10^{-5}$ | $1.368 \times 10^{-5}$ |
| Reflecting mirror 6 | $-1.631 \times 10^{-2}$ | $-1.087 \times 10^{-1}$ | $-1.555 \times 10^{-2}$ | $2.852 \times 10^{-7}$ | $2.787 \times 10^{-7}$ | $2.241 \times 10^{-7}$ |
| Reflecting mirror 7 | $-1.894 \times 10^{-2}$ | $-2.056 \times 10^{-2}$ | $-8.355 \times 10^{-2}$ | $2.591 \times 10^{-7}$ | $-3.888 \times 10^{-6}$ | $3.640 \times 10^{-7}$ |

After removing the rigid-body displacements of the reflecting surfaces, Zernike polynomials were utilized to fit the optical surface waves and obtain the Zernike polynomial coefficients. The results were then imported into the optical design software along with

the rigid-body displacements to obtain the analysis results of the optical paths of the transmission, receiving, and aft optical units.

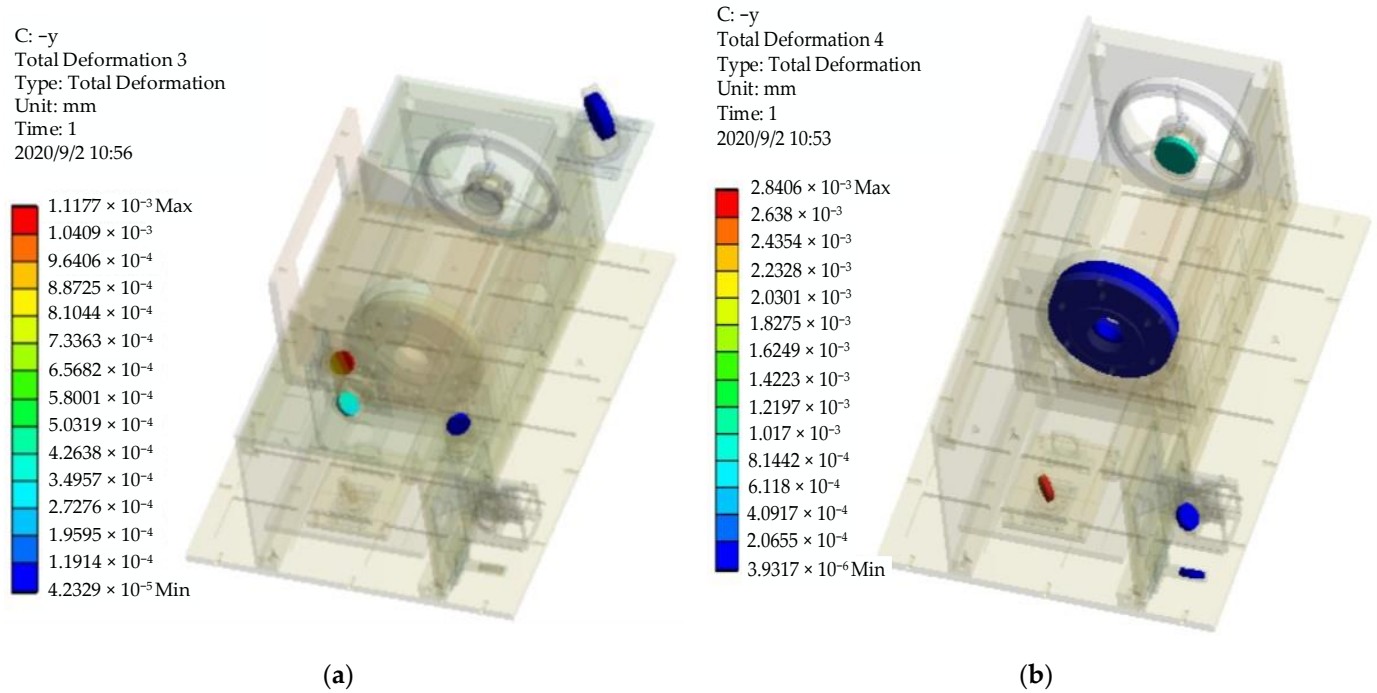

(**a**)　　　　　　　　　　　　　　　　　　(**b**)

**Figure 5.** Cloud diagram of reflecting mirror deformation under a gravity load: (**a**) transmission unit; (**b**) receiving and aft optical units.

### 3.2.1. Transmission Unit

With the automatic adjustment frame reflecting mirror as the reference, the image plane was set 20 m outward. The dispersion spot corresponding to the transmission unit module was then obtained as shown in Figure 6.

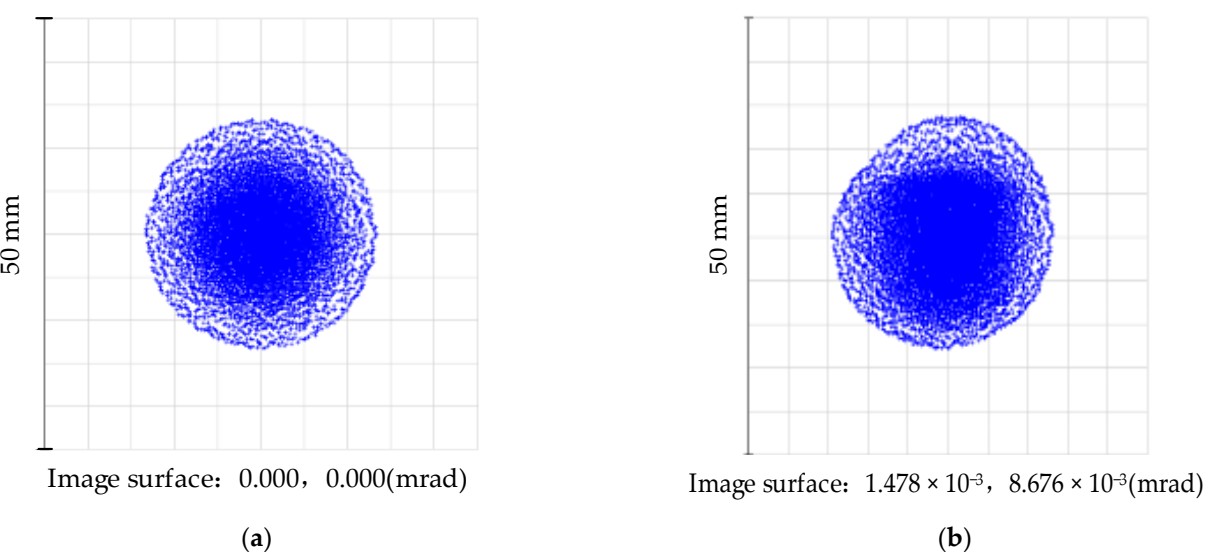

Image surface: 0.000，0.000(mrad)　　　Image surface: $1.478 \times 10^{-3}$，$8.676 \times 10^{-3}$(mrad)

(**a**)　　　　　　　　　　　　　　　　　　(**b**)

**Figure 6.** Dispersion spot of the transmission unit module: (**a**) no gravity load; (**b**) under a gravity load.

As shown in Figure 6, the outgoing beam of the transmission unit under a gravity load is deflected with a deflection angle of $8.8 \times 10^{-3}$ mrad and a divergence angle of

116 μrad, which is an increase of 8.6% compared to the gravity load-free value of 106 μrad. This is mainly caused by the gravity−induced microdeformation of the optical surface of reflecting mirror 1.

### 3.2.2. Receiving and Aft Optical Units

Taking the target surface of the detector as the study object, the receiving and aft optical unit optical paths in a field of view of 250 μrad (full angle) were simulated using parallel light with an incidence angle of ±125 μrad, and the detector target surface spot in the ±125 μrad field of view was calculated as shown in Figure 7.

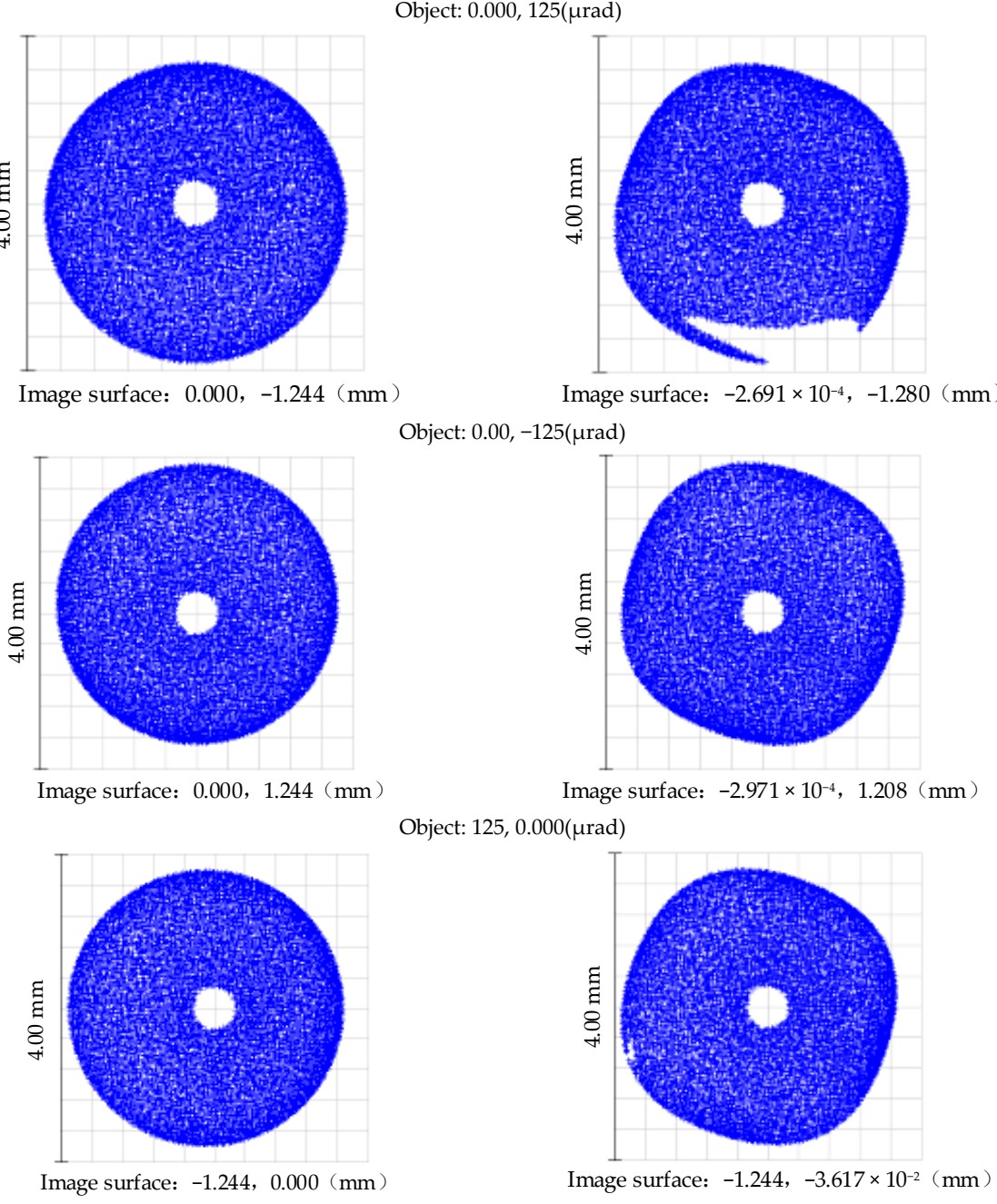

Object: 0.000, 125(μrad)

Image surface：0.000，−1.244（mm）　　Image surface：−2.691 × 10⁻⁴，−1.280（mm）

Object: 0.00, −125(μrad)

Image surface：0.000，1.244（mm）　　Image surface：−2.971 × 10⁻⁴，1.208（mm）

Object: 125, 0.000(μrad)

Image surface：−1.244，0.000（mm）　　Image surface：−1.244，−3.617 × 10⁻²（mm）

**Figure 7.** *Cont.*

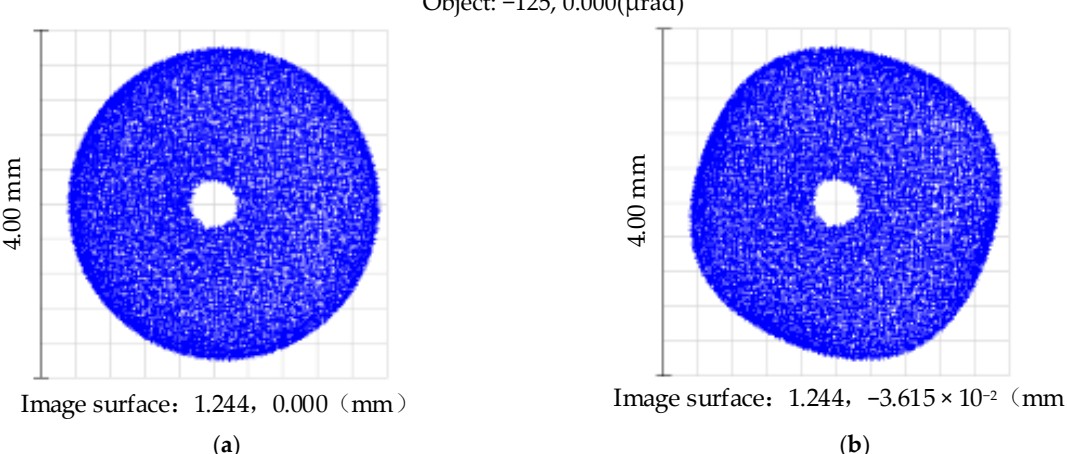

Object: −125, 0.000(μrad)

4.00 mm

4.00 mm

Image surface：1.244，0.000（mm）

Image surface：1.244，−3.615 × 10⁻² （mm）

(**a**)

(**b**)

**Figure 7.** Detector target surface spot in a ±125 μrad (half angle) field of view: (**a**) no gravity load; (**b**) under a gravity load.

The telescope field-of-view variation was further investigated as shown in Figure 7.

As shown in Figure 8, the radius of the dispersion spot at the diffraction limit is 12.361 μm when the encircled energy fraction is 86.5%. The maximum radius of the dispersion spot was 21.501 μm, and the diameter of the iris required was calculated to be 407.202 μm.

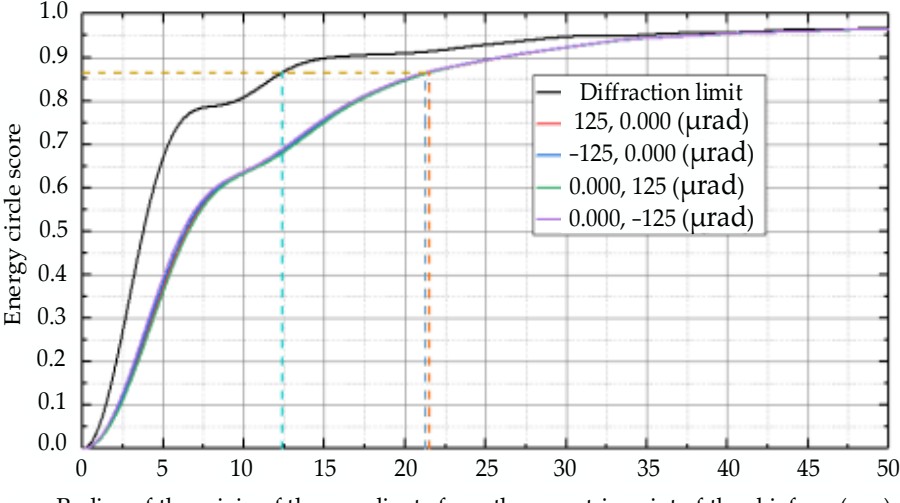

**Figure 8.** Energy concentration of the 125 μrad (half angle) telescope system under a gravity load.

In summary, with the simulated 125 μrad (half angle) field of view at different incidence angles, the eccentricity and radius of the dispersion spots are changed to different degrees, which is due to the relative displacement of the telescope's primary- and secondary-mirror optical surfaces and their surface deflections under the effect of gravity, resulting in a receiving field of view smaller than 250 μrad with a 0.4 mm iris. Simulation suggested that the focal length of the telescope increased by 89 μm, and the field of view of the telescope was 244 μrad, which was 2.4% less than 250 μrad.

## 4. LiDAR System Validation

### 4.1. Installation and Testing of the Transmission Unit

The optical–mechanical structure of the transmission unit was installed and tested. Figure 9a shows the actual picture of the installation and test experiment of the transmission unit. In the experiment, the laser and adjusted beam expander were first fixed on the

working plate. The direction of the laser beam from the laser was adjusted by the beam-steering structure and highly collimated into the beam expander, and then it was reflected out by the two mirrors.

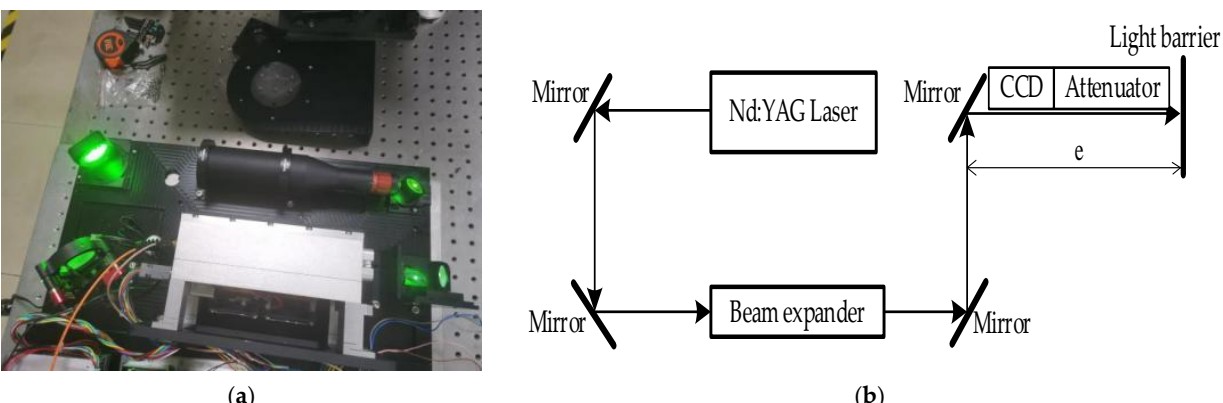

| (**a**) | (**b**) |

**Figure 9.** (**a**) Optical–mechanical structure of the transmission unit and (**b**) detection optical path of divergence angle.

To detect the divergence angle of the laser, the detection optical path was designed as shown in Figure 9b. The spot on the light barrier was detected using a charge-coupled device (CCD) camera, and the diameter of the spot with an encircled energy of 86.5% was selected. At e = 0.5 m, the detected average value of the spot diameter was 21.050 mm. At e = 20 m, the detected average value of the spot diameter was 21.270 mm, and the calculated divergence angle of the laser was 113 μrad, showing a relative error of 2.586% compared to the simulated value of 116 μrad under a gravity load, which was mainly determined using the actual divergence angle of the laser and the performance of the beam expander.

*4.2. Installation and Testing of the Receiving and Aft Optical Unit Optical–Mechanical Structures*

Figure 10 shows the actual setup for the installation and testing of the receiving and aft optical units. In the experiment, the receiving and aft optical units were first placed on the high-precision three-dimensional adjusting working platform, and the light source was then turned on. By adjusting the position of the receiving and aft optical unit structures, the parallel light emitted from the parallel-light tube was allowed to be normally incident on the system structure. The position of the iris was then adjusted using a high-precision three-dimensional adjusting structure to identify the focal position of the telescope system, and the eyepiece was adjusted to turn the beam into parallel light. Finally, the parallel light became normally incident on a narrowband filter under the effect of beam-steering mirrors, entered a polarizing prism, and split into two detection channels. The average value of the diameter of the detector target surface spot detected by the CCD camera was 3.860 mm, and the simulated value under a gravity load was 3.698 mm, displaying a relative error of 4.196%. Such a spot can be completely detected by a detector with a diameter of 8 mm.

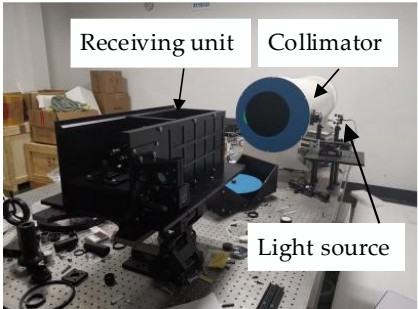

**Figure 10.** Installation and testing of the receiving and aft optical unit optical–mechanical structures.

### 4.3. Analysis of Continuous Detection Results

Continuous detection experiments of clouds and aerosols in the atmosphere over Hefei were conducted using the system from 09:00 a.m. on 2 November 2020 to 07:38 a.m. on 5 November 2020. The distance-corrected signals for the Parallel and Vertical polarization channels during the detection period are shown in Figure 11, with larger values representing higher aerosol content. The time evolution of the inverted depolarization ratio profiles of clouds and aerosols during the detection period is shown in Figure 12.

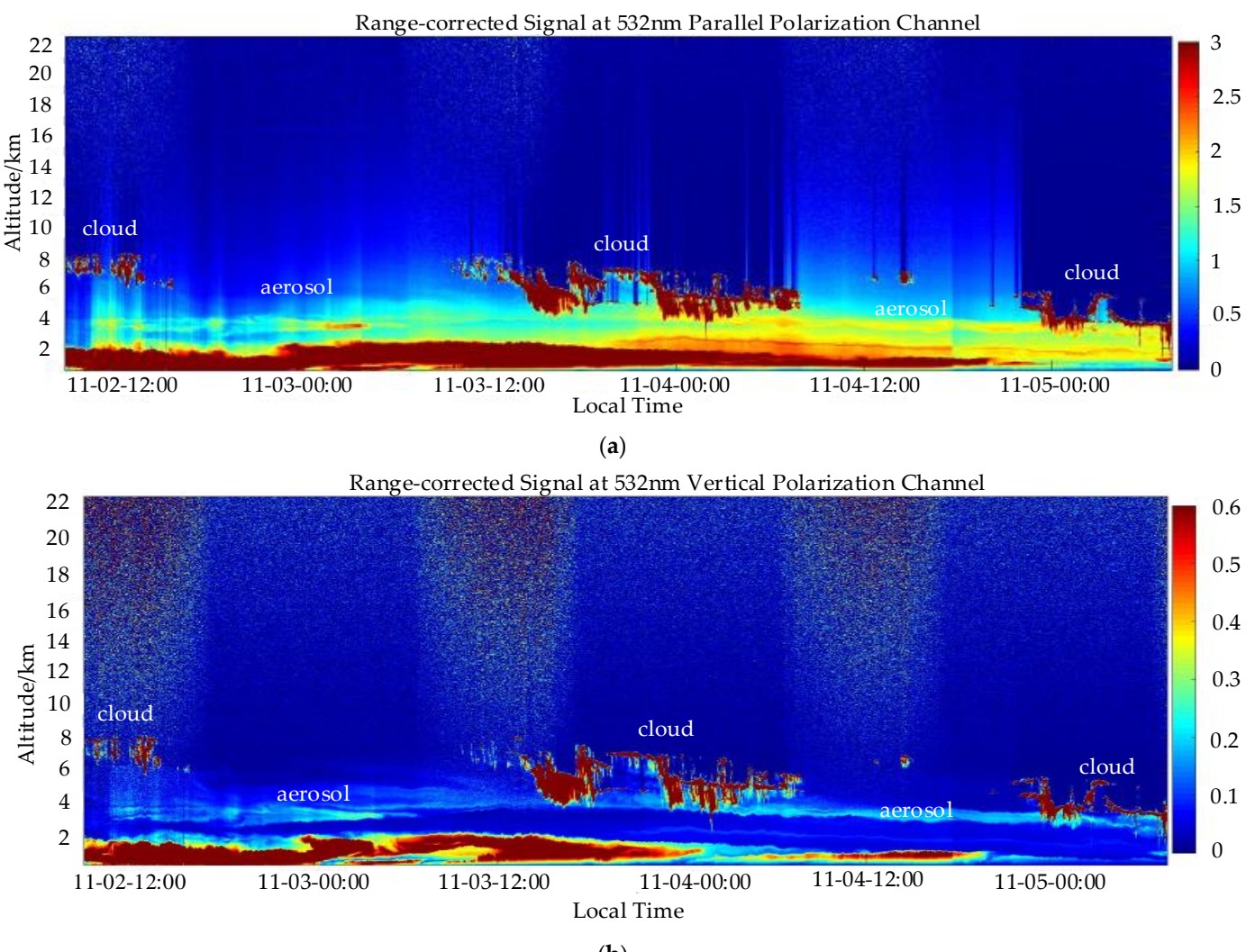

**Figure 11.** Distance-corrected signals for (**a**) Parallel and (**b**) Vertical Palarization channels during the continuous detection period from 2 November 2020 to 5 November 2020.

According to the results, the two channels completely recorded the spatial distribution of clouds and aerosols during the detection period, and more than 90% of the clouds and aerosols were found to be distributed below 10 km, i.e., within the troposphere. On 2 November, the aerosol was clearly divided into two structural layers. The aerosol layer appearing at 3–6 km had a relatively wide vertical coverage depth and a tendency to sink, with the depolarization ratio varying around 0.061. Meanwhile, the aerosol layer present at approximately 2 km was relatively stable, with the depolarization ratio varying around 0.101. On 3 November, the aerosol layer at 3–6 km was found to appear in two layers with a tendency to sink. The lower aerosol layer disappeared around 6 am, with a tendency to sink during the disappearing process, and the depolarization ratio varied around 0.061. The upper aerosol layer was at 4–5 km, and the depolarization ratio varied around 0.051.

The height of the aerosol layer around 2 km reached the maximum around 11:00, which may be related to the sinking and disappearance of the aerosol in the upper layer, while the depolarization ratio decreased and varied around 0.07. From 4 to 5 November, the aerosol layer at 4–5 km kept sinking, and the depolarization ratio varied around 0.05, while an aerosol layer appeared at approximately 3 km, with the depolarization ratio varying around 0.021. The height of the aerosol layer around 2 km decreased to below 1.500 km and reached the maximum of 1.500 km in the evening, with the depolarization ratio varying around 0.061. The variation in aerosol in the range of depolarization ratio from 0 to 0.11 indicates that aerosols are mainly composed of spherical particles during the detection period.

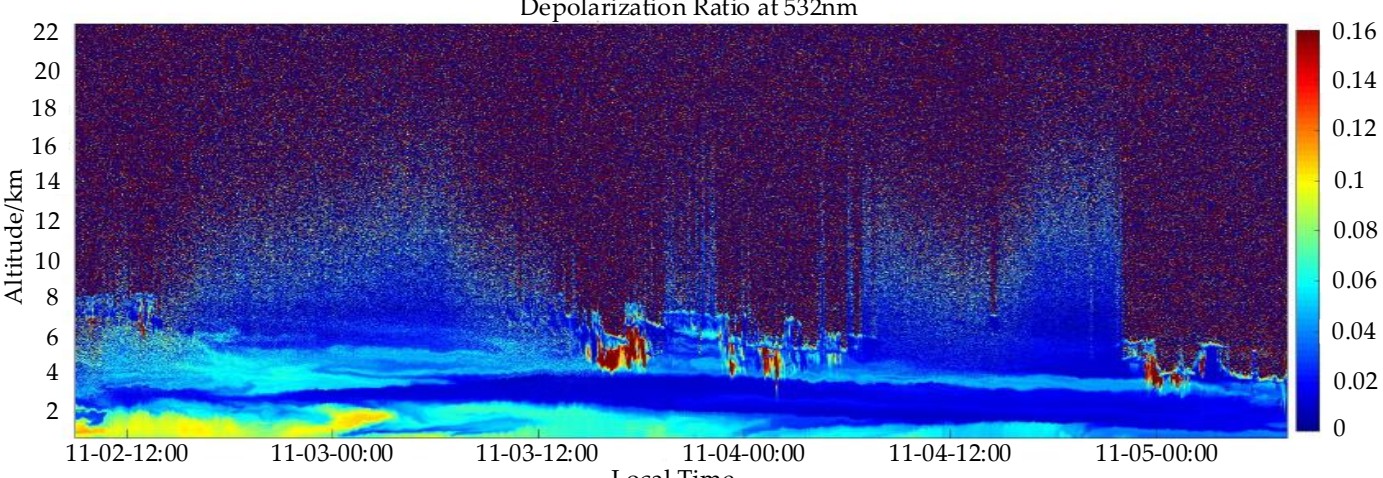

**Figure 12.** Inverted depolarization ratio profiles during continuous detection from 2 November 2020 to 5 November 2020.

During the detection period, clouds of different heights appeared in the time periods from 09:00 to 13:00 on 2 November, from 09:40 to 13:00 on 3 November, and from 22:30 on 4 November to 07:38 on 5 November. The depolarization ratio was greater than 0.4 when the cloud layer was at the same altitude as the aerosol layer and close to 0 when the cloud layer and aerosol layer were at different heights. This may be due to the interaction between the cloud and aerosol resulting in a larger depolarization ratio.

LiDAR was used to calculate the optical depth of aerosols, and to validate the calculation results, a comparative experiment was conducted with a sun photometer. Because the sun photometer is not accurate in the presence of clouds and can only be used for detection during daytime, the comparative experiment was conducted on 7 November 2020, when the sky was clear and cloudless. The corresponding results are shown in Figure 13.

LiDAR detection was conducted from 08:00 to 17:00 with a time resolution of 3 min, and the sun photometer detection was conducted from 09:00 to 16:30 with a time resolution of 1 min. The detection results of the two instruments were in good agreement, with maximum and minimum relative errors of 14.121% and 0.221%, respectively. The average values of the optical depth detected using LiDAR and the sun photometer were 0.314 and 0.329, respectively, with an average error of 4.559% throughout the day. This indicates that the designed LiDAR system can meet the requirements for the detection of optical depth of atmospheric aerosols.

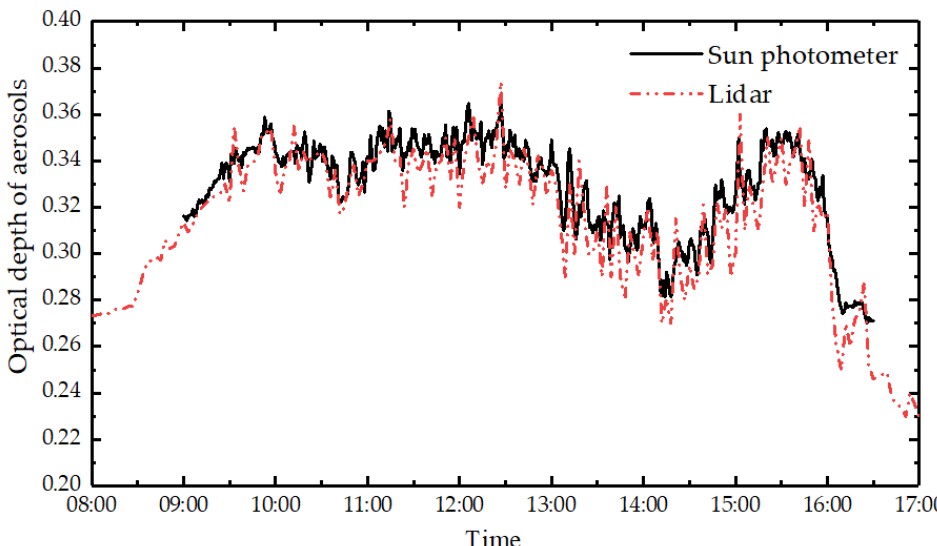

**Figure 13.** Comparison of inverted aerosol optical depths determined using LiDAR and the sun photometer on 7 November 2020.

## 5. Conclusions

The small-field-of-view LiDAR system with high repetition frequency, low energy, and single-photon detection technology serves as a data validation system for the development of spaceborne LiDAR in the later stage. Owing to the complexity of integrated systems, there has been no systematic analysis, and studies have neglected the influence of the structural micro-vibration of optical elements inside the optical system on the optical quality of the systems. Therefore, the Zernike polynomial fitting algorithm was programmed to realize transmission between optical and mechanical data, and optical–mechanical integration technology was employed to obtain the optical parameters of the integrated system under a gravity load in the process of designing the optical–mechanical structure of the integrated system. Compared to the experimental validation results, the divergence angle of the transmitting unit was 116 μrad according to optical–mechanical integration analysis, showing a relative error of 2.586%; the telescope focal length increased by 89 μm, the telescope field of view was 244 μrad, and the diameter of the detector target surface spot was 3.698 mm, with a relative error of 4.196%.

The continuous day/night detection results showed that the system could accurately detect the temporal and spatial variations in clouds and aerosols. The inverted optical depths were experimentally compared with those obtained using a solar photometer. The detection results of the two instruments were in good agreement, with maximum and minimum relative errors of 14.121% and 0.221%, respectively. The average optical depth was 0.314 and 0.329 as detected using LiDAR and the sun photometer, respectively, with an average detection error of 4.559%.

This indicates that the established optical mechanical integrated analysis model can conduct systematic analysis of LiDAR systems under external load conditions, and the analysis results are reliable and effective. This study provides technical support for further studying the impact of different operating conditions on the optical performance of the integrated LiDAR system, as well as optimizing the optical and mechanical structure design and thermal control of the spaceborne LiDAR system in the later stage.

**Author Contributions:** Conceptualization, resources, supervision, funding acquisition, writing—original draft preparation, L.L.; data curation, K.X.; software, B.W.; writing—review and editing, M.Z.; visualization, J.C. and P.Z. All authors have read and agreed to the published version of the manuscript.

**Funding:** This research was funded by the Key Project of Natural Science Research of Anhui Provincial Department of Education (No. KJ2021A0945); the high-level talent research start-up project of West Anhui University (No. WGK2022013); the Strategic Priority Research Program of the Chinese Academy of Sciences (No. XDA17040524); the Civil Aerospace Technology Advance Research Project (No. D040103); the Key Program of the 13th 5-year plan, CASHIPS (No. KP-2019-05); and the 2019 Anhui Province Science and Technology Major Project (No. 201903c08020013).

**Institutional Review Board Statement:** Not applicable.

**Informed Consent Statement:** Not applicable.

**Data Availability Statement:** The data presented in this study are available on request from the first author.

**Acknowledgments:** In this section, we would like to thank the Key Laboratory of Atmospheric Optics, the Anhui Institute of Optical and Fine Mechanics, Chinese Academy of Sciences, for providing funding.

**Conflicts of Interest:** The authors declare no conflicts of interest.

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
