# Peer review of "Optical–Mechanical Integration Analysis and Validation of LiDAR Integrated Systems with a Small Field of View and High Repetition Frequency"

_photonics, doi:10.3390/photonics11020179_

Round 1

Reviewer 1 Report

Comments and Suggestions for Authors

The paper presents an interesting application of optical-mechanical integration analysis to improve the design of a lidar system. The methodology seems sound, and the results demonstrate the value of this analysis approach. However, a couple of points need to be addressed:

1.      The literature review is too short and lacks comprehensiveness. In the introduction, provide a more substantial background on optical-mechanical integration analysis to better motivate its application in this context. Briefly explain how this analysis works, highlight the current state-of-the-art, and elucidate why the proposed integration is considered advanced.

2.      In Section 2.1, provide an explanation for the weighting factor (wi) in function (4).

3.      In Section 3.2, the description of the cloud diagrams is unclear. Consider revising this paragraph for clarity.

4.      Sections 3.2.1 and 3.2.2 contain several grammar issues that should be addressed.

5.      In Section 4.3, enhance the clarity of the explanation of the depolarization ratio results, particularly for non-experts.

6.      Strengthen the conclusion by highlighting the key outcomes and contributions more explicitly.

Comments on the Quality of English Language

as above. minor issues.

Author Response

Dear Reviewer.

Thanks very much for taking your time to review this manuscript. Please refer to the attachment for specific reply and modification.

Reviewer 2 Report

Comments and Suggestions for Authors

The authors introduce an opto-mechanical study of a micropulse LiDAR. The deformation induces by gravity is studied and consequences on the field shape and on the field of view of both the emission and reception are evaluated and compared to an experimental setup showing good agreement.

Specific comments:

line 83 : needed reference
line 99  : (-1)^(i) : more explanation is needed here
line 100: replace Table 1 by Table 2.1
line 102 : where is the definition of "design accuracy requirements"
line 112 : A space is missing between the two sentences
line 119 : D is not defined. Or rephrase sentence line 119.
line 123 : a conclusion on the paragraph and an introduction to the next paragraph would be welcomed
line 200 : typo : ridius --> Radius

General comments:

1. The quicklooks show data over the blind zone. No information is given on the overlap function of the LiDAR. Has it been evaluated with and without taking into account the gravity effect ?

2. Is it possible to push the study taking into account temperature effect?

Author Response

Dear Reviewer,

Thanks very much for taking your time to review this manuscript. Please refer to the attachment for specific reply and modification.

Reviewer 3 Report

Comments and Suggestions for Authors

This study takes a small-field-of-view LiDAR system with high repetition frequency, low energy, and single-photon detection technology as an example, programs the Zernike polynomial fitting algorithm to enable transmission between optical and mechanical data; and employs the optical–mechanical integration technology to obtain the optical parameters of the integrated system.

The major weaknesses are short of the innovation and the verifications using the various real environments.

Another weakness is English grammars and a few inappropriate terms and the sentences writing. Thus, a native speaker for polishing the English grammars is needed.

The references are heavily not sufficient, resulting in a short of overviewing the most current advances in the relevant field.

Comments on the Quality of English Language

poor

Author Response

(The authors gave the same response as above.)

Round 2

Reviewer 1 Report

Comments and Suggestions for Authors

Authors have solved all issues.